# Bounce Cosmology in Generalized Modified Gravities

**Georgios Minas [1], Emmanuel N. Saridakis [2,3]** 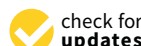**, Panayiotis C. Stavrinos [4,*]** **and Alkiviadis Triantafyllopoulos [1]**

[1] Section of Astrophysics, Astronomy and Mechanics, Department of Physics, National and Kapodistrian University of Athens, Panepistimiopolis, 15784 Athens, Greece; geminas@phys.uoa.gr (G.M.); alktrian@phys.uoa.gr (A.T.)

[2] Department of Physics, National Technical University of Athens, Zografou Campus, GR 157 73 Athens, Greece; msaridak@phys.uoa.gr

[3] Department of Astronomy, School of Physical Sciences, University of Science and Technology of China, Hefei 230026, China

[4] Department of Mathematics, National and Kapodistrian University of Athens, Panepistimiopolis, 15784 Athens, Greece

\* Correspondence: pstavrin@math.uoa.gr

**Abstract:** We investigate the bounce realization in the framework of generalized modified gravities arising from Finsler and Finsler-like geometries. In particular, a richer intrinsic geometrical structure is reflected in the appearance of extra degrees of freedom in the Friedmann equations that can drive the bounce. We examine various Finsler and Finsler-like constructions. In the cases of general very special relativity, as well as of Finsler-like gravity on the tangent bundle, we show that a bounce cannot easily be obtained. However, in the Finsler–Randers space, induced scalar anisotropy can fulfil bounce conditions, and bouncing solutions are easily obtained. Finally, for the general class of theories that include a nonlinear connection, a new scalar field is induced, leading to a scalar–tensor structure that can easily drive a bounce. These features reveal the capabilities of Finsler and Finsler-like geometries.

**Keywords:** bounce cosmology; Finsler geometry; modified gravity

## 1. Introduction

Bounce cosmologies offer an alternative view of the early universe [1–6] (for a review, see Reference [7]). Historically, this idea belongs to Tolman, who first suggested in the 1930s the possibility of a re-expansion of a closed universe that has already collapsed to an extremely dense state [8]. Since then, various bouncing models have been proposed in an effort for a systematic explanation of the origin of our universe.

The main advantage of bouncing cosmology is that it provides a way of solving the singularity problem that appears in the standard cosmological paradigm. The singularity (Big Bang) is replaced with a smooth transition from contraction to expansion (Big Bounce). In this sense, bounce cosmology offers the opportunity to obtain a more continuous picture of the early universe. The efficiency of bouncing models in solving basic cosmological problems in comparison with inflationary scenarios is visualized via the wedge diagram introduced in Reference [9].

In general, the realization of a bounce requires violation of the null-energy condition. This can be achieved with the introduction of extra degrees of freedom that are added ad hoc into the Lagrangian [4,10]. The violation of the null-energy condition needs to be handled with care, in order to not spoil the usual thermal history and the sequence of epochs after the bounce. Nevertheless, such violations can be easily acquired from modified [7] or quantum gravity [11]. In particular, they can easily be acquired, for example, in the Pre-Big Bang [12,13] and the Ekpyrotic [14,15] models, in gravity actions with higher-order corrections [1,16], in $f(R)$ gravity [17,18], in $f(T)$ gravity [19], in braneworld scenarios [20,21], in nonrelativistic gravity [22–24], in Galileon theory [25,26], in massive gravity [27], in Lagrange-modified gravity [28], and in loop quantum

cosmology [29–31]. Moreover, a nonsingular bounce model that supports magnetogenesis at the inflationary epoch is presented in Reference [32].

Among modified gravity theories, an interesting class is that of gravitational models based on Finsler and Finsler-like geometries. These are natural extensions of Riemannian geometry in which physical quantities may directly depend on observer four-velocity, and this velocity dependence reflects the Lorentz-violating character of the kinematics. Such a property is called *dynamic anisotropy* [33–44]. Additionally, Finsler and Finsler-like geometries are strongly connected to effective geometry within anisotropic media [45,46], and naturally enter the analog gravity program [47]. These features suggest that Finsler and Finsler-like geometries may play an important role within quantum gravity physics. The dependence of the metric tensor and other quantities on the position coordinates of the base manifold and the directional/velocity variables of the tangent space suggest that the natural geometrical framework for the description of these models is the tangent bundle of a smooth manifold. Finally, in the case where there is no velocity dependence, Finsler geometry becomes Riemannian.

The intrinsic geometrical spacetime dynamical anisotropy of Finsler geometry (not to be confused with the spatial anisotropy that may exist also in Riemannian geometry, as, for instance, in Bianchi cases) is included in the geometry of spacetime as an intrinsic field (variable) that influences its geometrical and physical concepts. Hence, it can give us the form of anisotropy as a hypothetical field, the *anisotropion*, which produces this deviation from isotropy. This appears in Friedmann equations and Lorentz violations [48–51], and thus anisotropy arises as a property of Finslerian spacetime [48,49,52,53].

In the present work, we are interested in investigating bounce realization in the framework of modified gravity related to Finsler and Finsler-like geometries. In particular, we desired to see how the new features of Finsler geometry can drive bouncing solutions, and to examine the evolution of intrinsic anisotropy during the bounce. In some bouncing scenarios, anisotropy decreases in the contracting phase and remains quite small during the bounce, in agreement with the current observational data [4]. On the other hand, there are also scenarios where anisotropy reduction in the contracting phase is followed by its exponential growth during the bounce, mainly due to the quantum fluctuations of the curvature [54]. Finally, we mention that nonsingular bounces are also possible to be generated in models, which spontaneously violate Lorentz symmetry [50,55,56]. In this framework, Lorentz symmetry violations lead to interactions with anisotropies [57]. Hence, we can establish a connection between anisotropic fields and a nonsingular bounce. In summary, we depict the above form of connections in the diagram of Figure 1.

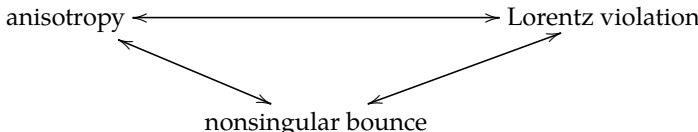

**Figure 1.** Connections of anisotropy, nonsingular bounce, and Lorentz violation.

The outline of this work is as follows: In Section 2, we first describe the basic conditions for a bounce realization and we briefly review Finsler geometry and gravity. Then, we examine bounce realization in general very special relativity and the Finsler–Randers models. In Section 3, we study the case of Finsler-like gravity on a tangent bundle, while in Section 4 we analyze bouncing solutions from scalar–tensor theory on the fiber bundle. Finally, in Section 5 we present a summary and our conclusions.

## 2. Bounce from Finsler Gravity

In this section, we study the bounce realization in the framework of Finsler gravity. We start by describing the conditions for bounce realization, and provide the basics of Finsler geometry and gravity. Then, we proceed to examine bounce realization in specific models, such as general very special relativity and Finsler–Randers models.

### 2.1. Bounce Conditions

Let us start by discussing the basic requirements for a bouncing solution. For the moment, we consider the ordinary Friedmann–Robertson–Walker (FRW) geometry with metric

$$[g_{\mu\nu}(x)] = \text{diag}\left(-1, \frac{a^2(t)}{1-kr^2}, a^2(t)r^2, a^2(t)r^2\sin^2\theta\right), \tag{1}$$

with $a(t)$ being the scale factor and $k = -1, 0, +1$ corresponding to open, flat, and closed spatial geometry, respectively. As usual, in such a geometry the general field equations of any theory give rise to the Friedmann and Raychaudhuri equations, which can be written in a compact form as:

$$H^2 = \frac{8\pi G}{3}\rho_{tot} - \frac{k}{a^2} \tag{2}$$

$$\dot{H} = -4\pi G(\rho_{tot} + P_{tot}) + \frac{k}{a^2}, \tag{3}$$

where $G$ is Newton's constant, $H = \dot{a}/a$ is the Hubble function, and with dots denoting derivatives with respect to cosmic time $t$. In the above expressions, $\rho_{tot}$ and $P_{tot}$ are, respectively, the total energy density and pressure of the universe, which include matter, radiation, dark energy, and any other gravitational or geometrical contribution that a theory or scenario may have.

In order to obtain a bounce realization, we need a contracting universe, namely, with $H < 0$, succeeded by an expanding universe, namely, with $H > 0$; hence, from continuity we deduce that, at the bounce point, we must have $H = 0$. Furthermore, one can see that, at the bounce point and around it, we must have $\dot{H} > 0$. Observing the form of the general Friedmann and Raychaudhuri Equations (2) and (3), and focusing on the physically more interesting flat case, we deduced that the above requirements could be fulfilled if

$$\rho_{tot} = 0 \tag{4}$$

exactly at the bounce point, and if additionally the null-energy condition is violated around the bounce point, namely, if

$$\rho_{tot} + P_{tot} < 0 \tag{5}$$

(in the case of a nonflat universe, the bounce can be driven by the curvature term without null-energy-condition violation [7]). Therefore, in order to obtain a bounce, one needs to construct theories in which the extra contributions to the total energy density and pressure are such that the null-energy condition is violated around the bounce point and the requirement of Equation (5) holds; moreover, total energy becomes zero exactly at the bounce point, and the condition of Equation (4) holds. As we see in the following, scenarios based on Finsler gravity can fulfil these necessary conditions.

### 2.2. Finsler Gravity

We first briefly review the basics of Finsler gravity, since this lies in the center of the investigation of the present work. Finsler gravity is a geometrical extension of general relativity, where the role of the metric is played by real-valued fundamental function $F(x, y)$, defined on tangent bundle $TM$ over a smooth spacetime manifold $M$. Variable $y$ is an element of the tangent space of $M$ at a point $x$ (we suppressed indices for convenience). The distance of two neighboring points on $M$ is defined as $ds = F(x, dx)$. We consider the following properties to hold:

1.  $F$ is continuous on $TM$, and smooth on $\widetilde{TM} \equiv TM \setminus \{0\}$, i.e., the tangent bundle minus the null section.

2.  $F$ is positively homogeneous of first degree on its second argument:

$$F(x, ky) = kF(x, y), \qquad k > 0. \tag{6}$$

3. Form

$$f_{\mu\nu}(x, y) = \frac{1}{2}\frac{\partial^2 F^2}{\partial y^\mu \partial y^\nu} \tag{7}$$

defines a nondegenerate matrix on $TM$ minus null set $\{(x, y) \in TM | F(x, y) = 0\}$:

$$\det\left[f_{\mu\nu}\right] \neq 0. \tag{8}$$

Using homogeneity condition Equation (6), it can be shown that:

$$F^2(x, y) = |f_{\mu\nu}(x, y)y^\mu y^\nu|; \tag{9}$$

therefore, $f_{\mu\nu}(x, y)$ can play the role of the metric for the vector space spanned by $y$. When studying gravity, metric $f_{\mu\nu}(x, y)$ is considered to be of Lorentzian signature $(-, +, +, +)$.

*2.3. General Very Special Relativity on Cosmology*

A particularly interesting Finslerian cosmological model is elaborated in the framework of the so-called general very special relativity on cosmology [57]. The metric function takes form

$$F(x, y) = \left(g_{\mu\nu}(x)y^\mu y^\nu\right)^{(1-b)/2}\left(n_\kappa y^\kappa\right)^b, \tag{10}$$

where $g_{\mu\nu}(x)$ is the ordinary FRW metric, Equation (1). Equation (10) is a direct cosmological generalization of the general very special relativity description, where the line element is

$$ds = \left(\eta_{\mu\nu}dx^\mu dx^\nu\right)^{(1-b)/2}\left(n_\kappa dx^\kappa\right)^b, \tag{11}$$

with $[\eta_{\mu\nu}] = \mathrm{diag}(-1, 1, 1, 1)$, which is invariant under transformations generated by deformation $DISIM_b(2)$ of Lorentz subgroup $ISIM(2)$ [58,59]. One-form $n_\kappa$ is called a "spurionic field". We mention that parameter $b$ quantifies the deviation from Riemannian geometry, i.e., the Lorentz violation in the gravitational sector. Parameterized post-Newtonian (PPN) analysis [60] and the use of solar-system data provide the most stringent constraints on it; thus, Gravity Probe B puts an upper bound at $10^{-7}$ [61].

The Riemannian osculating approach is followed, namely, $g_{\mu\nu}(x) = f_{\mu\nu}(x, y(x))$, where $y(x)$ is the tangent vector to the cosmological fluid's (matter fluid) flow lines. As usual, the matter fluid is described by the energy–momentum tensor of the perfect fluid:

$$T_{\mu\nu} = P_m g_{\mu\nu} + (\rho_m + P_m)y_\mu y_\nu, \tag{12}$$

where $\rho_m$ is the energy density and $P_m$ the pressure. The field equations for this construction are then:

$$L_{\mu\nu} - \frac{1}{2}L g_{\mu\nu} = -8\pi G T_{\mu\nu}, \tag{13}$$

where $L_{\mu\nu}$ is the Ricci tensor for metric $g_{\mu\nu}(x)$ and $L = g^{\mu\nu}L_{\mu\nu}$.

Applying the above geometrical construction in a cosmological framework, we considered the spurionic field to be parallel to the velocity of the comoving observer, namely,

$$n^\kappa = \left(n(t), 0, 0, 0\right). \tag{14}$$

As a simple model, in Reference [57], we imposed the following approximations

$$n(t) \approx At + B$$
$$A \to 0 \tag{15}$$
$$B \to 0,$$

since $n(t)$, parameterized by $A,B$, needs to be suitably small in order to be consistent with the observational small bound on $b$. For these choices, the Ricci tensor components for the metric function, Equation (10), are calculated as [57]:

$$L_{00} = 3\frac{\ddot{a}}{a} + 3\frac{Ab}{B}\frac{\dot{a}}{a} + O\left(A^2\right)$$

$$L_{11} = -\frac{a\dot{a} + 2\dot{a}^2 + 2k}{1 - kr^2} + \frac{5A}{B}b\frac{a\dot{a}}{1 - kr^2} + O(A^2)$$

$$L_{22} = -r^2(a\ddot{a} + 2\dot{a}^2 + 2k) - \frac{5A}{B}br^2a\ddot{a} + O(A^2)$$

$$L_{33} = -r^2(a\ddot{a} + 2\dot{a}^2 + 2k)\sin^2\theta - \frac{5A}{B}br^2a\ddot{a}\sin^2\theta$$

$$+O(A^2).$$

(16)

Therefore, using the above, we obtain the following generalization of the Friedmann equations:

$$H^2 + \frac{k}{a^2} + 2\frac{A}{B}bH = \frac{8\pi G}{3}\left[\rho_m - 2\frac{A}{B}bP_m\left(t + \frac{B}{A}\ln B\right)\right]$$

(17)

$$\dot{H} + H^2 + \frac{Ab}{B}H = -\frac{4\pi G}{3}\left[(\rho_m + 3P_m)\right.$$

$$\left. + 4\ln(At + B)b(\rho_m + P_m)\right].$$

(18)

Unfortunately, as one can see, the above Friedmann equations do not accept a bounce solution. One could still try to construct a model with a different approximation than Equation (16) of Reference [57], but such a detailed investigation of a new construction lies beyond the scope of the present work. Hence, in the following subsection, we examine the case of another Finslerian construction, where bounce realization is possible.

### 2.4. Bounce in Finsler–Randers Space

Let us now consider a different Finslerian construction, namely, Finsler–Randers (FR) space [62,63]. In this space, a Lagrangian metric function is given by

$$F(x, y) = \alpha(x, y) + u_\mu y^\mu, \quad \|u_\mu\| \ll 1,$$

(19)

where $\alpha(x, y) = \sqrt{g_{\kappa\lambda}(x)y^\kappa y^\lambda}$, and $g_{\kappa\lambda}(x)$ is the FRW metric, Equation (1), with $\kappa, \lambda, \mu \in \{0, 1, 2, 3\}$.

In this cosmological model, an important role is played by the variation of anisotropy $Z_t$. In the case of the FRW geometry, Equation (1), the modified Friedmann equations of the generalized form of FR-type cosmology were studied in Reference [63], and are written as:

$$H^2 = \frac{8\pi G}{3}\rho_m - HZ_t - \frac{k}{a^2},$$

(20)

$$\dot{H} = -4\pi G\left(\rho_m + P_m\right) + \frac{1}{4}HZ_t + \frac{k}{a^2}.$$

(21)

In these expressions, we defined the variation of anisotropy $Z_t$ as $Z_t = \dot{u}_0$ as the derivative of the time component of unit vector $\hat{u}_a$ [63]. This variation affects the form of geometry, as can be seen from Equations (20) and (21), and, at the limit $Z_t \to 0$, we recovered the ordinary Friedmann equations of general relativity. Finally, we considered the matter sector to correspond to a perfect fluid with energy density and pressure $\rho_m$ and $P_m$, respectively.

Observing the form of two Friedmann Equations (20) and (21), we can define the effective energy density and pressure of geometrical origin as

$$\rho_{FR} \equiv -\frac{3}{8\pi G} H Z_t \tag{22}$$

$$P_{FR} \equiv \frac{5}{16\pi G} H Z_t. \tag{23}$$

Therefore, total energy density and pressure, respectively, become $\rho_{tot} = \rho_m + \rho_{FR}$ and $P_{tot} = P_m + P_{FR}$, and the Friedmann equations take the usual form of Equations (2) and (3). Hence, we can now easily examine what the conditions are in order to fulfil the bounce requirements of Equations (4) and (5).

First, from Equation (4) we deduce that a flat universe exactly at bounce point $\rho_m$ must be zero ($\rho_{FR}$ also becomes zero exactly at the bounce point since $H = 0$). This is a usual assumption in many bouncing models, and it is expected to be fulfilled in the early universe. Taking this into account, we moreover see that the condition of Equation (5) implies that, around the bouncing point, $\rho_{FR} + P_{FR} < 0$ and, thus, that $H Z_t > 0$. Hence, we deduce that the above requirements can be fulfilled if we suitably choose variation of anisotropy $Z_t$.

In order to provide a specific example, we focused on a flat FRW geometry ($k = 0$), and we considered a bouncing scale factor of the form

$$a(t) = a_b (1 + Bt^2)^{1/3}, \tag{24}$$

where $a_b$ is the scale factor value at the bounce, while $B$ is a positive parameter that determines how fast the bounce takes place. In this case, time varies between $-\infty$ and $+\infty$, with $t = 0$ being the bouncing point, and where, away from the bounce, one obtains the usual expansion behavior. Moreover, we considered that the matter sector is absent in the early universe. Inserting these into Equation (20), we immediately find that

$$Z_t = -\frac{2Bt}{3(1 + Bt^2)}. \tag{25}$$

Hence, it is this $Z_t$, which comes from the Finslerian modification of the geometry, that generates bouncing-scale factor of Equation (24). Moreover, we remark that variation of anisotropy $Z_t$ actually determines physically important quantity $B$ in Equation (24).

## 3. Finsler-Like Gravity on a Tangent Bundle

In this Section, we are interested in examining whether a bounce can be realized from Finsler-like gravity on a tangent bundle. Generally, we use the term Finsler-like for any metric theory in which the various structures may depend on a set of internal variables ($y, \phi$, etc) apart from the position or external ones, which we denote as $x^\mu$ through this work. Finsler-like extensions of general relativity on the tangent bundle are presented in the bibliography [64–68], and bouncing cosmological scenarios were studied on them [49,69,70]. In the following, we focus our interest on a tangent bundle $TM$ equipped with a Finslerian Sasaki-type metric:

$$\mathcal{G} = g_{\mu\nu}(x, y)\, dx^\mu \otimes dx^\nu + v_{\alpha\beta}(x, y)\, \delta y^\alpha \otimes \delta y^\beta, \tag{26}$$

where $x^\mu$ are the coordinates on the base manifold, with $\kappa, \lambda, \mu, \nu, \ldots = 0, 1, 2, 3$, and $y^\alpha$ are the fiber coordinates, with $\alpha, \beta, \ldots, \theta = 0, 1, 2, 3$. On the total space $TTM$ of $TM$, the adapted basis is $\{\delta_\mu, \dot{\partial}_\alpha\}$, and its dual is given by $\{dx^\mu, \delta y^\alpha\}$. The following definitions hold:

$$\delta_\mu = \frac{\delta}{\delta x^\mu} = \frac{\partial}{\partial x^\mu} - N_\mu^\alpha(x, y)\frac{\partial}{\partial y^\alpha}$$

$$\dot{\partial}_\alpha = \frac{\partial}{\partial y^\alpha}$$

$$\delta y^\alpha = dy^\alpha + N_\nu^\alpha dx^\nu, \tag{27}$$

where $N_\mu^\alpha(x, y)$ are the coefficients of a nonlinear connection on $TM$. This connection is defined by a splitting of the total space $TTM$ of $TM$ into an h-subspace $HTM$ spanned by $\{\delta_\mu\}$, and a v-subspace $VTM$ spanned by $\{\dot\partial_\alpha\}$ [64]. The tangent space of $TM$ is thus a Whitney sum of the h-subspace and v-subspace, namely,

$$TTM = HTM \oplus VTM. \tag{28}$$

One can now introduce $d-$connection $\mathcal{D}$ as a covariant linear differentiation rule that preserves h-space and v-space:

$$\mathcal{D}_{\delta_\kappa}\delta_\nu = L_{\nu\kappa}^\mu(x, y)\delta_\mu \qquad \mathcal{D}_{\dot\partial_\gamma}\delta_\nu = C_{\nu\gamma}^\mu(x, y)\delta_\mu \tag{29}$$

$$\mathcal{D}_{\delta_\kappa}\dot\partial_\beta = L_{\beta\kappa}^\alpha(x, y)\dot\partial_\alpha \qquad \mathcal{D}_{\dot\partial_\gamma}\dot\partial_\beta = C_{\beta\gamma}^\alpha(x, y)\dot\partial_\alpha. \tag{30}$$

A canonical $d-$connection is a linear connection that is compatible with metric of Equation (26), and it preserves, under parallel translation, horizontal and vertical subspaces $HTM$ and $VTM$ [64]. It can be uniquely defined if one demands that it only depends on $g_{\mu\nu}$, $v_{\alpha\beta}$ and $N_\mu^\alpha$, and moreover that connection coefficients $L_{\nu\kappa}^\mu$ and $C_{\beta\gamma}^\alpha$ are symmetric on the lower indices. In this case, its coefficients turn out to be [71]:

$$L_{\nu\kappa}^\mu = \frac{1}{2}g^{\mu\rho}\left(\delta_k g_{\rho\nu} + \delta_\nu g_{\rho\kappa} - \delta_\rho g_{\nu\kappa}\right)$$

$$L_{\beta\kappa}^\alpha = \dot\partial_\beta N_\kappa^\alpha + \frac{1}{2}v^{\alpha\gamma}\left(\delta_\kappa v_{\beta\gamma} - v_{\delta\gamma}\dot\partial_\beta N_\kappa^\delta - v_{\beta\delta}\dot\partial_\gamma N_\kappa^\delta\right)$$

$$C_{\nu\gamma}^\mu = \frac{1}{2}g^{\mu\rho}\dot\partial_\gamma g_{\rho\nu}$$

$$C_{\beta\gamma}^\alpha = \frac{1}{2}v^{\alpha\delta}\left(\dot\partial_\gamma h_{\delta\beta} + \dot\partial_\beta h_{\delta\gamma} - \dot\partial_\delta v_{\beta\gamma}\right). \tag{31}$$

Now, the curvature of the nonlinear connection is defined as

$$\Omega_{\nu\kappa}^\alpha = \frac{\delta N_\nu^\alpha}{\delta x^\kappa} - \frac{\delta N_\kappa^\alpha}{\delta x^\nu}, \tag{32}$$

and the space at hand is equipped with various Ricci curvature tensors such as:

$$\overline{R}_{\mu\nu} = \delta_\kappa L_{\mu\nu}^\kappa - \delta_\nu L_{\mu\kappa}^\kappa + L_{\mu\nu}^\rho L_{\rho\kappa}^\kappa - L_{\mu\kappa}^\rho L_{\rho\nu}^\kappa \tag{33}$$

$$S_{\alpha\beta} = \dot\partial_\gamma C_{\alpha\beta}^\gamma - \dot\partial_\beta C_{\alpha\gamma}^\gamma + C_{\alpha\beta}^\epsilon C_{\epsilon\gamma}^\gamma - C_{\alpha\gamma}^\epsilon C_{\epsilon\beta}^\gamma. \tag{34}$$

Hence, the generalized Ricci scalar curvature reads as

$$\mathcal{R} = g^{\mu\nu}\overline{R}_{\mu\nu} + v^{\alpha\beta}S_{\alpha\beta} \equiv \overline{R} + S. \tag{35}$$

One can now write a Hilbert-like action, namely [64–67],

$$\begin{aligned}\mathcal{S}_{TM} &= \frac{1}{16\pi G}\mathcal{S}_H + \mathcal{S}_M \\ &\equiv \frac{1}{16\pi G}\int d^8\mathcal{U}\sqrt{\det\mathcal{G}}\,\mathcal{L}_H + \int d^8\mathcal{U}\sqrt{\det\mathcal{G}}\,\mathcal{L}_M,\end{aligned} \tag{36}$$

with

$$d^8\mathcal{U} = dx^0 \wedge dx^1 \wedge dx^2 \wedge dx^3 \wedge dy^0 \wedge dy^1 \wedge dy^2 \wedge dy^3, \tag{37}$$

where the gravitational part of action $\mathcal{S}_H$ is constructed by gravitational Lagrangian

$$\mathcal{L}_H = \mathcal{R} = (\overline{R} + S), \tag{38}$$

and matter action $\mathcal{S}_M$ by matter Lagrangian $\mathcal{L}_M$.

Extremization of total action $\mathcal{S}_{TM}$ with respect to metric components $g_{\mu\nu}$ and $v_{\alpha\beta}$ leads to the following field equations [49]:

$$\overline{R}_{(\mu\nu)} - \frac{1}{2}(\overline{R} + S)g_{\mu\nu} = 8\pi G T_{\mu\nu} \tag{39}$$

$$S_{\alpha\beta} - \frac{1}{2}(\overline{R} + S)v_{\alpha\beta} = 8\pi G Y_{\alpha\beta}, \tag{40}$$

where we defined $T_{\mu\nu} = -\frac{2}{\sqrt{\det\mathcal{G}}}\frac{\delta(\mathcal{L}_M\sqrt{\det\mathcal{G}})}{\delta g^{\mu\nu}}$ and $Y_{\alpha\beta} = -\frac{2}{\sqrt{\det\mathcal{G}}}\frac{\delta(\mathcal{L}_M\sqrt{\det\mathcal{G}})}{\delta v^{\alpha\beta}}$. Applying these field equations in the FRW metric Equation (1), focusing on the flat case and, assuming usual matter perfect fluid Equation (12), one obtains the following modified Friedmann equations [49]:

$$H^2 = \frac{8\pi G}{3}\rho_m - \frac{1}{6}S \tag{41}$$

$$\dot{H} + H^2 = -\frac{4\pi G}{3}\left(\rho_m + 3P_m\right) - \frac{1}{6}S, \tag{42}$$

where, due to the imposed symmetries, all quantities only depend on time.

From the form of the two Friedmann Equations (41) and (42), we can see that we obtained extra contributions that reflect the Finsler-like structure of the tangent bundle. In particular, these induce effective energy density and pressure of geometrical origin as

$$\rho_S \equiv -\frac{1}{16\pi G}S \tag{43}$$

$$P_S \equiv \frac{1}{16\pi G}S. \tag{44}$$

Hence, total energy density and pressure, respectively, become $\rho_{tot} = \rho_m + \rho_S$ and $P_{tot} = P_m + P_S$, and the Friedmann equations acquire the usual form of Equations (2) and (3). Thus, we can examine what the conditions are in order to fulfil the bounce requirements of Equations (4) and (5). Concerning Equation (4), we deduced that, for a flat universe exactly at the bounce point, we must have $S = 16\pi G\rho_m$, while Equation (5) requires $\rho_m + P_m < 0$ (since, according to Equations (43) and (44), $P_S + \rho_S = 0$). Therefore, we conclude that, in the case of a flat universe and for standard matter, a bounce cannot be obtained in the scenario at hand.

Nevertheless, a bounce could still be possible with the addition of extra fields, e.g., Reference [49], but one still has to be careful with the constraints imposed to $S$ via Equation (40). For example, if we consider the trivial case where $Y_{\alpha\beta} = 0$, then the trace of Equation (40) gives

$$S = -2\overline{R}. \tag{45}$$

We assume that the extra field can be modeled to a perfect fluid as in Equation (12), with energy density and pressure $\rho_{eff}$ and $P_{eff}$, respectively; thus, Friedmann Equation (41) takes the form

$$H^2 = \frac{8\pi G}{3}(\rho_m + \rho_{eff}) - \frac{1}{6}S. \tag{46}$$

Substituting Equation (45) to Equation (46)[1] gives $3H^2 + 2\dot{H} + 8\pi G(\rho_m + \rho_{eff})/3 = 0$. This relation implies that, in order for an extra field with trivial $Y_{\alpha\beta}$ to induce a bounce solution for our spatially flat metric, it would need to have $\rho_{eff} < 0$, which is undesirable from a physical point of view.

---

[1]  In our case, $\overline{R}$ reduces to the ordinary flat FRW Ricci scalar curvature of general relativity due to the fact that metric components $g_{\mu\nu}(x)$ do not depend on $y$ variables, as was shown in Reference [49].

## 4. Bounce from Scalar–Tensor Theory on the Fiber Bundle

In this section, we investigate bounce generation in theories that include scalar–tensor sectors on the fiber bundle. These constructions are very general, with a very rich structure and behavior, which reveals the significant capabilities of Finsler-like geometry. We first present the basics of this construction, and then we proceed to the investigation of two explicit scenarios.

### 4.1. Model

We consider a fibered space over a pseudo-Riemannian spacetime manifold $M$ of the form $M \times \{\phi^{(1)}\} \times \{\phi^{(2)}\}$, where $\phi^{(1)}, \phi^{(2)}$ stand for the fiber coordinates. Under coordinate transformations on the base manifold, fiber coordinates behave like scalars. Moreover, the space is equipped with a nonlinear connection with coefficients $N_\mu^{(\alpha)}(x^\nu, \phi^{(\beta)})$, where $\mu, \nu$ take values from 0 to 3, and $\alpha, \beta$ take the values 1 and 2 [52]. Its adapted bases for the tangent and cotangent spaces are $\{\delta_\mu = \partial_\mu - N_\mu^{(\beta)}\partial_{\phi^{(\beta)}}, \partial_{\phi^{(\alpha)}}\}$, where a summation is implied over the possible values of $\beta$, and $\{dx^\mu, \delta\phi^{(\alpha)} = d\phi^{(\alpha)} + N_\mu^{(\alpha)}dx^\mu\}$ with a summation implied over the possible values of $\mu$. The metric structure of the space is defined as [52]:

$$\mathbf{G} = g_{\mu\nu}(x)\, dx^\mu \otimes dx^\nu + v_{(\alpha)(\beta)}(x)\, \delta\phi^{(\alpha)} \otimes \delta\phi^{(\beta)}. \tag{47}$$

The metric coefficients for the fiber coordinates are set as $v_{(0)(0)} = v_{(1)(1)} = \phi(x^\mu)$ and $v_{(0)(1)} = v_{(1)(0)} = 0$. Note that function $\phi$ is clearly a scalar under coordinate transformations. A detailed investigation of the above construction was performed in Reference [52], where a metrical d-connection was introduced and its curvature and torsion tensor coefficients were calculated. Additionally, the Raychaudhuri equations for the model were derived in Reference [53].

We can now write an action as [53]:

$$\mathcal{S}_G = \frac{1}{16\pi G} \int \sqrt{|\det \mathbf{G}|}\, \mathcal{L}_G dx^{(N)}, \tag{48}$$

where $\mathcal{L}_G$ is taken equal to the scalar curvature of the d-connection, and $dx^{(N)} = d^4x \wedge d\phi^{(1)} \wedge d\phi^{(2)}$. In the special case of a holonomic basis, i.e., $[\delta_\mu, \delta_\nu] = 0$, the scalar curvature of the d-connection is

$$\mathcal{R} = R - \frac{2}{\phi}\Box\phi + \frac{1}{4\phi^2}\partial^\mu\phi\partial_\mu\phi, \tag{49}$$

where $R$ is the scalar curvature of the Levi-Civita connection, and $\Box$ is the d'Alembert operator with respect to it. On the other hand, in the general case, one obtains the scalar curvature as

$$\tilde{\mathcal{R}} = R - \frac{2}{\phi}\Box\phi + \frac{1}{4\phi^2}\partial^\mu\phi\partial_\mu\phi + \frac{1}{\phi}\partial^\mu\phi\,\partial_{\phi^{(\alpha)}}N_\mu^{(\alpha)}. \tag{50}$$

Additionally, we can add the matter sector, too, considering total action

$$\mathcal{S} = \frac{1}{16\pi G} \int \sqrt{|\det \mathbf{G}|}\, \mathcal{L}_G dx^{(N)} + \int \sqrt{|\det \mathbf{G}|}\, \mathcal{L}_M dx^{(N)}. \tag{51}$$

Since, for determinants $\det \mathbf{G}$ and $\det g$ ,we have relation $\det \mathbf{G} = \phi^2 \det g$, the above total action can be rewritten as

$$\mathcal{S} = \frac{1}{16\pi G} \int \sqrt{|\det g|}\, \phi\mathcal{L}_G dx^{(N)} + \int \sqrt{|\det g|}\, \phi\mathcal{L}_M dx^{(N)}. \tag{52}$$

In the following two subsections, we separately study the bounce realization in the holonomic ($\mathcal{L}_G = \mathcal{R}$) and nonholonomic ($\mathcal{L}_G = \tilde{\mathcal{R}}$) basis.

### 4.2. Bounce in Holonomic Basis

Let us consider the total action Equation (52) in the case of the holonomic basis, also allowing for a potential for the scalar field, namely [53],

$$
\mathcal{S} = \frac{1}{16\pi G} \int \sqrt{|\det g|}\left[\phi\mathcal{R} - V(\phi)\right]dx^{(N)}
$$

$$
+ \int \sqrt{|\det g|}\,\phi\mathcal{L}_M dx^{(N)},
\tag{53}
$$

where $\mathcal{R}$ is the holonomic scalar curvature of Equation (49). We mention here that the above action belongs to the Horndeski class; hence, the resulting equations of motion are guaranteed to have up to second-order derivatives [72]. In particular, the field equations for the metric are extracted as

$$
E_{\mu\nu} = 8\pi G T_{\mu\nu} + \frac{1}{\phi}\left(\nabla_\mu\nabla_\nu\phi - g_{\mu\nu}\Box\phi\right)
$$

$$
+ \frac{1}{4\phi^2}\left[\frac{1}{2}g_{\mu\nu}(\nabla\phi)^2 - \nabla_\mu\phi\nabla_\nu\phi\right] - \frac{1}{2\phi}g_{\mu\nu}V,
\tag{54}
$$

where $E_{\mu\nu} = R_{\mu\nu} - \frac{1}{2}Rg_{\mu\nu}$ is the Einstein tensor, $T_{\mu\nu} = -\frac{2}{\sqrt{|g|}}\frac{\delta(\sqrt{|g|}\mathcal{L}_M)}{\delta g^{\mu\nu}}$ is the energy–momentum tensor, and $\nabla_\mu$ is the Levi-Civita covariant derivative, while the scalar-field (extension of Klein–Gordon) equation reads as

$$
\Box\phi = 2\phi\left(R - V'\right) + \frac{1}{2\phi}(\nabla\phi)^2 + 32\pi G\mathcal{L}_M\phi,
\tag{55}
$$

with $V' = dV/d\phi$. It is interesting to note that, in the scenario at hand, we obtained effective interaction between the scalar field and the matter sector due to the transformation from a **G**-metric to a $g$-metric.

Applying the above equations to the FRW metric Equation (1), focusing on the flat case, and neglecting the matter sector, since we are interested in early-time bounce realization, we obtained the following modified Friedmann equations:

$$
3H^2 = -3H\frac{\dot\phi}{\phi} - \frac{\dot\phi^2}{8\phi^2} + \frac{1}{2\phi}V
\tag{56}
$$

$$
\dot H + H^2 = -\frac{1}{2\phi}\left(\ddot\phi + H\dot\phi\right) + \frac{\dot\phi^2}{12\phi^2} + \frac{V}{6\phi}
\tag{57}
$$

$$
\ddot\phi + 3H\dot\phi = -12\phi\left(2H^2 + \dot H\right) + \frac{\dot\phi^2}{2\phi} + 2\phi V',
\tag{58}
$$

out of which two are independent.

We now proceed to show how it is possible to obtain a specific bounce in this construction. As we observed from the above equations, we may choose specific scalar-field potential that can satisfy the general bounce conditions of Equations (4) and (5) and, thus, induce bounce realization. We follow the procedure of References [19,26–28,73], and we first start from the desired result, that is, we impose a known form of scale factor $a(t)$ possessing bouncing behavior. Thus, $H(t)$ is known, too. Eliminating $V$ from Equations (56) and (57) gives simple differential equation

$$
4\phi(t)\ddot\phi(t) - \dot\phi(t)[\dot\phi(t) + 4H(t)\phi(t)] + 8\dot H(t)\phi(t)^2 = 0,
\tag{59}
$$

which can be solved to provide $\phi(t)$. Then, this $\phi(t)$ can be inserted into Equation (56) and provide $V(t)$ as

$$
V(t) = 6H(t)[\dot\phi(t) + \phi(t)H(t)] + \frac{\dot\phi(t)^2}{4\phi(t)}.
\tag{60}
$$

Finally, knowing both $\phi(t)$ and $V(t)$, eliminating time we can extract the explicit form of potential $V(\phi)$. Hence, it is this potential that generates the initially given desired bouncing scale factor $a(t)$.

Let us provide an explicit example of the bounce realization. We start by inserting desired bouncing scale factor of Equation (24) and we apply the above steps. Since analytical solutions cannot be obtained, we numerically solve Equation (59) and find $\phi(t)$, and then we use Equation (60) to find $V(t)$. These two functions are shown in Figure 2. Hence, from these $\phi(t)$ and $V(t)$, we reconstruct potential $V(\phi)$, which is depicted in Figure 3.

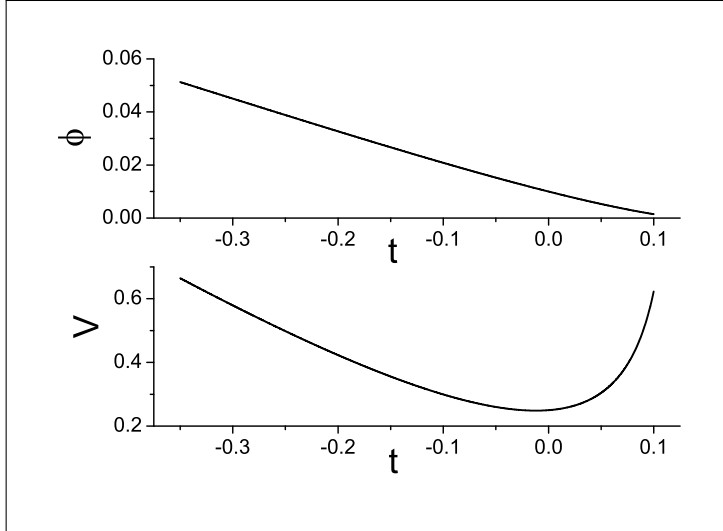

**Figure 2.** Solution for scalar field $\phi(t)$ (**upper graph**) and of potential $V(t)$ (**lower graph**), for the holonomic basis, under imposed bouncing scale factor Equation (24) with $B = 1$, in units where $8\pi G = 1$.

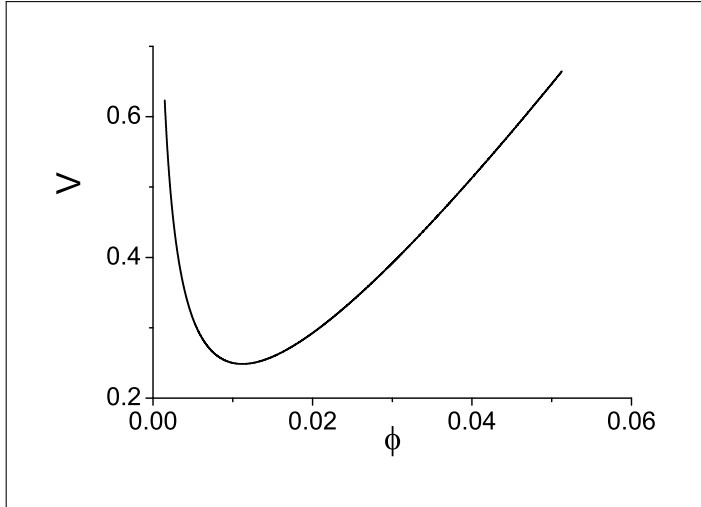

**Figure 3.** Reconstructed scalar potential $V(\phi)$ using Figure 2, under imposed bouncing scale factor Equation (24) with $B = 1$, in units where $8\pi G = 1$.

Therefore, if this $V(\phi)$ is imposed as an input, one acquires the bounce realization and, in particular, bouncing scale factor Equation (24).

### 4.3. Bounce in Nonholonomic Basis

We now proceed to the investigation of the nonholonomic case, namely, we consider the total action Equation (52) with $\mathcal{L}_G = \tilde{\mathcal{R}}$, i.e.,

$$S = \frac{1}{16\pi G} \int \sqrt{|g|}\, \phi \tilde{\mathcal{R}} dx^{(N)} + \int \sqrt{|g|}\, \phi \mathcal{L}_M dx^{(N)}, \tag{61}$$

where $\tilde{\mathcal{R}}$ is the nonholonomic scalar curvature of Equation (50). This action leads to the following equations of motion for the metric and the scalar field:

$$E_{\mu\nu} = 8\pi G T_{\mu\nu} + \frac{1}{\phi}\left(\nabla_\mu \nabla_\nu \phi - g_{\mu\nu} \Box \phi\right)$$

$$+ \frac{1}{4\phi^2}\left[\frac{1}{2} g_{\mu\nu}(\nabla\phi)^2 - \nabla_\mu \phi \nabla_\nu \phi\right]$$

$$- \left(\delta_\mu^\lambda \partial_\nu \phi - \frac{1}{2} g_{\mu\nu} \partial^\lambda \phi\right) N_\lambda \tag{62}$$

$$\Box\phi = 2\phi R + \frac{1}{2\phi}(\nabla\phi)^2 + 32\pi G \mathcal{L}_M \phi - \phi D^\mu N_\mu, \tag{63}$$

where $N_\mu \equiv \partial_{\phi^{(\alpha)}} N_\mu^{(\alpha)}$, and with $D_\mu N^\lambda = \delta_\mu N^\lambda + \Gamma^\lambda_{\kappa\mu} N^\kappa$ being the d-covariant differentiation on the fiber bundle where $\Gamma^\lambda_{\kappa\mu}$ are the Christoffel symbols. We note that the last term in Equation (63), which reflects the internal structure of Finsler-like geometry, can be seen to act as an effective potential for scalar field $\phi$. Since every other quantity in Equations (62) and (63) only depends on $x^\mu$ coordinates, this should also be the case for $N_\lambda$ for consistency (equivalently $\partial_{\phi^{(\beta)}} \partial_{\phi^{(\alpha)}} N_\mu^{(\alpha)} = 0$ on shell).

Applying the above equations of motion in FRW metric Equation (1), focusing on the flat case, and neglecting the matter sector, since we are interested in early-time bounce realization, leads to the modified Friedmann equations

$$3H^2 = -3H\frac{\dot\phi}{\phi} - \frac{\dot\phi^2}{8\phi^2} - \frac{1}{2}\dot\phi N_0 \tag{64}$$

$$\dot{H} + H^2 = -\frac{1}{2\phi}\left(\ddot\phi + H\dot\phi\right) + \frac{\dot\phi^2}{12\phi^2} + \frac{1}{3}\dot\phi N_0 \tag{65}$$

$$\ddot\phi + 3H\dot\phi = -12\phi\left(2H^2 + \dot{H}\right) + \frac{\dot\phi^2}{2\phi} + \phi\left(\dot{N}^0 + 3HN^0\right), \tag{66}$$

out of which two are independent, where, as we mentioned, due to symmetries, all quantities only depend on time. Thus, in the Friedmann equations, we acquire a modification reflecting the nonholonomicity of the fiber bundle of the underlying Finsler-like geometry.

Let us now show how this construction may give rise to bounce realization. From the form of Friedmann Equations (64) and (65), we deduce that we may choose a specific nonholonomic function $N^0(t)$ that can satisfy the general bounce conditions of Equations (4) and (5) and, thus, induce the bounce. We first start from the desired result, that is, we impose as input a scale factor form $a(t)$ that possesses bouncing behavior. Therefore, $H(t)$ is known, too. Eliminating $N^0$ from Equations (64) and (65) gives simple differential equation

$$\ddot\phi(t) + 5H(t)\dot\phi(t) + 2\phi(t)[\dot{H}(t) + 3H(t)^2] = 0, \tag{67}$$

which can be solved to provide $\phi(t)$. Then, this $\phi(t)$ can be substituted into Equation (64) and provide $N^0(t)$ as

$$N_0(t) = -6 \left[ \frac{H(t)}{\phi(t)} + \frac{\dot{\phi}(t)}{24\phi(t)^2} + \frac{H(t)^2}{\dot{\phi}(t)} \right]. \tag{68}$$

Hence, it is this $N_0(t)$, induced by the nonlinear connection of Finsler-like geometry, that generates the initially given desired bouncing scale factor $a(t)$.

We close this subsection by providing an explicit example of bounce realization. We use bouncing scale factor Equation (24) as input, and we apply the above steps. We numerically solve Equation (67) and find $\phi(t)$, and then use Equation (68) to find $N_0(t)$. In Figure 4, we depict the solution for $N_0(t)$. Hence, if this $N_0$ is imposed as input, one obtains the bounce realization and, in particular, bouncing scale factor Equation (24).

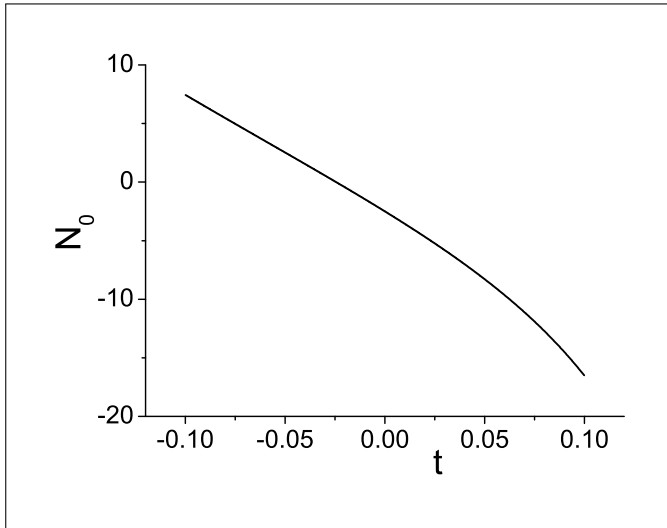

**Figure 4.** Reconstructed time-dependent part $N_0(t)$ related to the nonlinear connection for the nonholonomic basis under imposed bouncing scale factor Equation (24) with $B = 1$, in units where $8\pi G = 1$.

## 5. Conclusions

In this work, we investigated bounce realization in the framework of Finsler and Finsler-like gravity. Finsler and Finsler-like geometries are natural extensions of Riemannian geometry, where one allows that physical quantities may directly depend on observer four-velocity. Hence, gravitational theory based on Finsler and Finsler-like gravity provides gravitational modification, since it induces extra terms in the field equations. When applied to a cosmological framework, the richer intrinsic structure of Finsler and Finsler-like geometries is reflected in extra terms in the resulting modified Friedmann equations. Thus, these terms can lead to bounce realizations.

In our analysis, we considered various Finsler and Finsler-like constructions, and we examined whether bouncing solutions could be obtained. As a first model, we considered the so-called general very special relativity, which presents a slight Lorentz violation quantified by a single parameter, and the "spurionic" one form. As we showed, under linear approximation, this scenario cannot lead to a bounce. However, considering the Finsler–Randers space, in which the intrinsic Finslerian structure is reflected in the appearance of a new function in the Friedmann equations (the variation of anisotropy), we saw that bounce conditions could easily be fulfilled and, thus, the bounce could be realized.

As a next construction, we examined Finsler-like gravity on the tangent bundle. Performing analysis and considering the two involved curvature tensors, we extracted the Friedmann equations that contain a modification resulting from the tangent-bundle-related $S$-curvature. Nevertheless, for simple models and

standard matter, these extra terms cannot drive a bouncing solution since they cannot lead to violation of the null-energy condition.

As a last construction, we considered theories that include scalar–tensor sectors on the fiber bundle. These theories present a very rich structure revealing the capabilities of Finsler-like geometry. In particular, the nonlinear connection induces a new degree of freedom that behaves as a scalar under coordinate transformations. In a cosmological framework, this scalar field appears in the Friedmann equations, and therefore its dynamics may trigger a bounce. In the case of a holonomic basis, we showed that the bounce could easily be obtained, and we provided a way of the reconstruction of the potential that gives rise to a desired bouncing scale factor. Similarly, in the case of a nonholonomic basis, we saw that the bounce could easily be realized, and we presented the reconstruction procedure of the time coefficient related to the nonlinear connection that induces the desired bounce.

In summary, we saw that Finsler and Finsler-like geometries are natural frameworks for the realization of bounce cosmology. Apart from background evolution, one should additionally investigate various scenarios at the perturbation levels, since the process of perturbations through the bounce phase is strongly related to the subsequent development of the large-scale structure, and hence to observations. Such detailed perturbation analysis lies beyond the scope of the present work and it is left for future investigation.

**Author Contributions:** All authors have contributed equivalently.

**Funding:** This research is cofinanced by Greece and the European Union (European Social Fund (ESF)) through the Operational Program "Human Resources Development, Education and Lifelong Learning" in the context of the project "Strengthening Human Resources Research Potential via Doctorate Research" (MIS-5000432), implemented by the State Scholarships Foundation (IKY).

**Acknowledgments:** This article is based upon work from COST (European Cooperation in Science and Technology) Action CA15117 "Cosmology and Astrophysics Network for Theoretical Advances and Training Actions" (CANTATA).

**Conflicts of Interest:** The authors declare no conflict of interest. The funders had no role in the design of the study; in the collection, analyses, or interpretation of data; in the writing of the manuscript, or in the decision to publish the results.

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
