# Peer review of "Bounce Cosmology in Generalized Modified Gravities"

_universe, doi:10.3390/universe5030074_

Round 1

Reviewer 1 Report

This paper is correct, contains interesting results, and should be published.
In my opinion, however, there are a few missing references on bouncing models, Finsler-like geometries and broken Lorentz symmetry which may be of some interest (and some relevance) for the main topics of this paper, and which should be included.

In particular, for what concerns the pre-big bang scenario, the possibility of specific bouncing models has been discussed not in Ref. [13], but in the following references:

- M.~Gasperini, M.~Giovannini and G.~Veneziano,
  ``Perturbations in a nonsingular bouncing universe,''
  Phys.\ Lett.\ B {\bf 569}, 113 (2003)
  doi:10.1016/j.physletb.2003.07.028
  [hep-th/0306113].

- M.~Gasperini, M.~Giovannini and G.~Veneziano,
  ``Cosmological perturbations across a curvature bounce,''
  Nucl.\ Phys.\ B {\bf 694}, 206 (2004)
  doi:10.1016/j.nuclphysb.2004.06.020
  [hep-th/0401112].

In addition, a Finsler-like modification of general relativity, based on a direct dependence of the effective space-time metric on the four-velocity of a geodesic observer, was also proposed in the following paper:

- E.~R.~Caianiello, A.~Feoli, M.~Gasperini and G.~Scarpetta,
``Quantum Corrections to the Space-time Metric From Geometric Phase Space Quantization,'' Int.\ J.\ Theor.\ Phys.\  {\bf 29}, 131 (1990).
 doi:10.1007/BF00671323

and a class of bouncing cosmologies based on the above type of geometry was discussed in

- E. R. Caianiello, M. Gasperini and G. Scarpetta,
"Inflation and singularity prevention in a model for extended-object-dominated cosmology", Class. Quantum Grav. 8, 659 (1991)
https://iopscience.iop.org/article/10.1088/0264-9381/8/4/011/meta

- M.~Gasperini,
 ``A Geometric regularization procedure for the curvature of cosmological background", published in "Advances in Theoretical Physics - Italo-Soviet Workshop" (World Scientific, 1991), p. 77.
https://doi.org/10.1142/1449

Finally, non-singular bounces due to a breaking of the Lorentz symmetry where first considered in

- M.~Gasperini,
  ``Inflation and Broken Lorentz Symmetry in the Very Early Universe,''
  Phys.\ Lett.\  {\bf 163B}, 84 (1985).
  doi:10.1016/0370-2693(85)90197-2

- M.~Gasperini,
 ``Repulsive gravity in the very early universe,''
  Gen.\ Rel.\ Grav.\  {\bf 30}, 1703 (1998)
  doi:10.1023/A:1026606925857
  [gr-qc/9805060].

My suggestion is that the authors should consider the possibility of completing their final list if references by including the above mentioned papers. After such a small improvement, this paper may be accepted for publication in "Universe".

Author Response

REFEREE 1 GENERAL COMMENT:
This paper is correct, contains interesting results, and should be published.

AUTHORS:
We thank the referee for his/her report on our manuscript aiming at
improving the quality of the paper.  

REFEREE 1 POINT-1:
In my opinion, however, there are a few missing references on bouncing models,
Finsler-like geometries and broken Lorentz symmetry which may be of some interest (and
some relevance) for the main topics of this paper, and which should be included. 

In particular, for what concerns the pre-big bang scenario, the possibility of specific
bouncing models has been discussed not in Ref. [13], but in the following references:

AUTHORS:
The referee is correct in his point. We apologize for missing these relevant references.
In the revised version we have added them. Moreover, we have removed the old reference
[13] since it is indeed not related to the analysis.

Reviewer 2 Report

The authors work on cosmological implications of Finsler geometry. Their approach is particular interesting and by utilizing the formalism of Finsler geometry they derive the Friedmann equations and realize certain classes of cosmological bounces. The article contains interesting material and also pedagogically written parts, containing mathematical information on the fibre bundle structure of Finsler geometry. The article is well written, and I recommend publication. Before acceptance I recommend that the authors put frames in the figures and also what are the time units in all their plots? 

Author Response

The authors work on cosmological implications of Finsler geometry. Their approach is
particular interesting and by utilizing the formalism of Finsler geometry they derive the
Friedmann equations and realize certain classes of cosmological bounces. The article
contains interesting material and also pedagogically written parts, containing
mathematical information on the fibre bundle structure of Finsler geometry. The article is
well written, and I recommend publication.

AUTHORS:
We thank the referee for his/her positive report on our manuscript.

REFEREE 2 POINT-1:
Before acceptance I recommend that the authors
put frames in the figures and also what are the time units in all their plots? 

AUTHORS:
In the figures we use units in which 8Ď€G=1. As we mention, the
time-scale of the figure (i.e. of the bounce) is determined by the values of B. In
these specific examples we take B=1 in units in which 8Ď€G=1, however one can use
any B value he desires and hence obtain the evolution time-scale he desires.

AUTHORS FINAL COMMENT:
We thank again the referee for his/her useful comments and suggestions. Let us believe
that the revised version is now suitable for publication.